# Early detection of clinically significant prostate cancer: protocol summary and statistical analysis plan for the ProScreen randomised trial

Jaakko Nevalainen ,[1] Jani Raitanen,[2] Kari Natunen,[1] Tuomas Kilpeläinen,[3,4] Antti Rannikko,[3,4] Teuvo Tammela,[1,5] Anssi Auvinen[1]

¹Tampere University, Tampere, Finland
²Faculty of Social Sciences (Health Sciences) and Gerontology Research Center, Tampere University, Tampere, Finland
³University of Helsinki, Helsinki, Finland
⁴Helsinki University Central Hospital, Helsinki, Finland
⁵Tampere University Hospital, Tampere, Finland

**Correspondence to**
Professor Jaakko Nevalainen;
jaakko.nevalainen@tuni.fi

## ABSTRACT

**Introduction** Evidence on the effectiveness of prostate cancer screening based on prostate-specific antigen is inconclusive and suggests a questionable balance between benefits and harms due to overdiagnosis, and complications from biopsies and overtreatment. However, diagnostic accuracy studies have shown that detection of clinically insignificant prostate cancer can be reduced by MRI combined with targeted biopsies.
The aim of the paper is to describe the analysis of the ProScreen randomised trial to assess the performance of the novel screening algorithm in terms of the primary outcome, prostate cancer mortality and secondary outcomes as intermediate indicators of screening benefits and harms of screening.

**Methods** The trial aims to recruit at least 111 000 men to achieve sufficient statistical power for the primary outcome. Men will be allocated in a 1:3 ratio to the screening and control arms. Interim analysis is planned at 10 years of follow-up, and the final analysis at 15 years. Difference between the trial arms in prostate cancer mortality will be assessed by Gray's test using intention-to-screen analysis of randomised men. Secondary outcomes will be the incidence of prostate cancer by disease aggressiveness, progression to advanced prostate cancer, death due to any cause and cost-effectiveness of screening.

**Ethics and dissemination** The trial protocol was reviewed by the ethical committee of the Helsinki University Hospital (2910/2017). Results will be disseminated through publications in international peer-reviewed journals and at scientific meetings.

**Trial registration number** NCT03423303

## STRENGTHS AND LIMITATIONS OF THIS STUDY

⇒ This population-based, randomised multicentre trial targeting at recruiting 111 000 men will provide high-quality evidence on the effectiveness of a novel screening strategy for prostate cancer mortality.
⇒ Broad eligibility criteria and pragmatic approach embedded in normal clinical practice enhance the external validity of the trial and provide evidence applicable to decision-making in public health and healthcare.
⇒ Challenges for the trial include the maintenance of high compliance to screening and the extent of opportunistic prostate-specific antigen testing in the population.

## INTRODUCTION

Prostate cancer (PCa) is the most common cancer in men in many industrialised countries and causes substantial mortality.[1] Screening based on blood prostate-specific antigen (PSA) has been shown to decrease PCa mortality, but the evidence from randomised trials is not conclusive.[2 3] Systematic reviews of randomised controlled trials have concluded that PSA screening may at best lower PCa mortality, but not all-cause mortality. However, the balance between benefits and harms was regarded as problematic due to frequent overdiagnosis, and complications from biopsies and overtreatment.[4–6]

Several studies have shown that detection of clinically insignificant PCa can be reduced by MRI combined with targeted biopsies of the suspect foci, instead of systematic biopsies of the entire prostate.[7 8] However, previous studies have mostly focused on the diagnostic performance, that is, cancer detection at a single evaluation. A hybrid screening/diagnostic study and a screening trial using MRI were recently published.[9 10]

Here, we describe the analysis of the ProScreen randomised screening trial to assess the performance of a novel screening algorithm in terms of the primary outcome, PCa mortality and secondary outcomes, used as intermediate indicators of benefits and harms of screening. Following good statistical practice, this statistical analysis plan (V.1.1) was finalised prior to completion of recruitment and short-term follow-up data collection. It was written following the guidelines provided in Gamble *et al*[11] as applicable. Any

unforeseen deviations from the plan will be described and justified carefully in the respective reports.

## TRIAL OVERVIEW
### Trial design
The ProScreen trial is a population-based, randomised multicentre trial that investigates the effectiveness of a novel screening strategy combining PSA, a four-kallikrein panel and MRI on PCa mortality over a 15-year period from randomisation.[12] The rationale is to minimise detection of clinically insignificant cancers, while maintaining a high sensitivity for aggressive cancers in order to reduce overdiagnosis without compromising mortality benefits. An interim analysis of PCa mortality is planned at 10 years of follow-up.

### Ethics and dissemination
On 15 January 2018, the trial was registered at ClinicalTrials.gov (NCT03423303). The ethical committee of Helsinki University Hospital reviewed the protocol (tracking no 2910/2017). Permission to collect data from healthcare registers was obtained from the Finnish Institute for Health and Welfare (before the era of FinData, Dnro THL/676/5.05.00/2018). Written informed consent is provided by each participant in the screening arm. Results will be disseminated through publications in international peer-reviewed journals and at scientific meetings.

Recruitment started in October 2018 and is still ongoing.

### Patient and public involvement
Patients or the public were not involved in the design, or conduct, or reporting, or dissemination plans of our research.

### Study population
All men aged 50–63 years (at the time of sampling of the trial population) with Finnish or Swedish as mother tongue residing in the trial municipalities constitute the trial population. Men with a prevalent PCa will be identified through the Finnish Cancer Registry or hospital pathology databases and excluded.

We have identified for the trial the entire target population from the Digital and Population Data Services Agency, comprehensively without any sampling. The initial trial population consists of men residing in Helsinki and Tampere.

Currently, we are increasing the sample size by recruiting men also in the other municipalities within the Helsinki and Tampere metropolitan areas (Vantaa, Espoo and Kauniainen, as well as Nokia, Lempäälä, Pirkkala, Ylöjärvi and Kangasala with a total of 57 000 men in the target age group). The target population covers comprehensively all eligible men in the municipalities, both in the original Helsinki and Tampere areas and the new municipalities.

### Sample size
We estimated that we could find 110 000–120 000 men in the target age group based on the population projections from 2020 to 2034 from the 10 municipalities.[13] We requested the overall number of deaths and the number of PCa deaths from Statistics Finland by age group from 1990 to 2019. The proportion of PCa deaths had barely changed at all during the 30-year period; hence, we based our sample size calculation on these figures. With a 1:3 random allocation to the screening arm relative to the control arm, we expected that we would follow up on 82 500 men in the control arm. Based on population statistics and causes of death registries, for the first 10 years of the trial, we estimated PCa mortality to be 50 deaths per 100 000 person-years. For the entire 15-year follow-up, our estimate was 67 deaths per 100 000 person-years. Given deaths due to other causes, we thus expected to include at least 240 and 520 PCa deaths in the control arm during the first 10 and 15 years of the trial, respectively.

Considering that a PSA-based screening could result into a 20% reduction in PCa mortality,[2] and that the addition of MRI and 4K to the screening protocol could improve the ability to detect clinically significant PCa,[14] we assumed a relative hazard of 0.75 for the screening relative to the control arm. Schoenfeld's formula indicates that an 80% power would be reached by a total of 506 PCa deaths[15] with type I error rate set at 5%. Assuming a total of 650 PCa deaths—520 in the control arm and 130 in the screening arm—the power of the study would be 89%. Hence, we aim at a final sample size of at least 111 000 men to ensure adequate statistical power and precision at the final analysis.

### Randomisation and screening intervals
All eligible men will be randomly allocated to screening and control arms in a 1:3 ratio. The rationale for the unequal randomisation arose from the logistics in the conduct of the trial: we first estimated the maximum number of men who could be screened with existing resources. Given that the size of the control arm that does not receive any intervention has hardly any resource implications, we decided on an allocation ratio that would yield a manageable size of the intervention arm. Hence, the underlying rationale was to fix the size of the screening and maximise the size of the control arm to optimise the power of the study given the context (source population).

Within the screening arm, rescreening interval is adapted by the baseline PSA:
► Men with initial PSA ≥3 ng/mL are re-invited every 2 years,
► Men with PSA 1.5–2.99 ng/mL every 4 years.
► Men with PSA <1.5 ng/mL after 6 years.

The control arm is unaware of being part of the trial, as they are not contacted or subject to any trial procedures. At PCa diagnosis, the aim is that all men in the control arm will also answer questionnaires, but otherwise they are treated according to standard clinical practice according to the well-established national treatment guidelines throughout the trial.

By the time of writing this plan, we have randomised 61 193 men with 15 299 allocated to the screening arm

and 45 894 to the control arm. Analyses will compare the entire screening arm, regardless of the actual screening attendance and interval employed, to the control arm, unless otherwise specified.

Randomisation list consists of batches of randomised men, that is, randomisation list is generated in parts to ensure that the time between randomisation and the invitation to the trial would not be excessive. Stratified randomisation was not considered necessary. The list is generated centrally by a designated study biostatistician at the coordinating unit, which maintains the documentation including programme codes and the resulting lists include information of randomisation dates, personal identification numbers (linkable to study ID number) and the arm allocated. Randomisation lists are only shared confidentially to study personnel if needed for study conduct.

## Screening procedures

At every screening attendance, three consecutive tests are conducted in a stepwise manner before biopsy:

1. All participating men give a blood sample for determination of PSA at a local laboratory.
2. If the PSA is 3 ng/mL or higher, a four-kallikrein panel is analysed from a second vial of plasma from the initial draw using an algorithm incorporating four proteins (total PSA, free PSA, intact PSA and human kallikrein-2) and age. The result is expressed as probability of a clinically significant PCa.
3. Men with both PSA ≥3 and kallikrein score ≥7.5% are referred to MRI. T2-weighted, diffusion-weighted and dynamic contrast-enhanced imaging is employed in accordance with the European Society for Urogenital Radiology guideline.[16] The findings are classified according to the Prostate Imaging Reporting and Data System (PI-RADS V.2.1), which is a 5-point scale to combine the MRI findings and indicate the likelihood of a significant cancer. Scores of 3–5 indicate at least a suspect finding warranting directed biopsy.

Only targeted biopsies are employed, with two to four cores per region of interest depending on the size. Only screen-positive men with negative MRI but PSA density >0.15 undergo systematic biopsy as a safety measure (to avoid missing clinically significant cancers). Similar fusion-guided biopsy systems are used at the two trial sites and evaluated by experienced uropathologists using standardised procedures.

A random sample of screen-negative (on test steps 1 and 2) men are also invited to prostate MRI and asked to give blood, urine and stool samples in order to serve as a control group to estimate frequency of suspicious MRI findings in the general population and as a reference group in analyses of biological samples.

## Protocol deviations

A tabular presentation of different types of protocol deviations along with their frequencies and percentages will be presented. Any protocol deviations detected after

randomisation will be carefully documented. Among them, men later found out not to have met the eligibility criteria at the date of randomisation will be excluded from the analysis (post-randomisation exclusions).

In the case of major protocol violations affecting a substantial proportion of men, separate per-protocol analyses will be conducted to support the main analyses. In the screening arm, incomplete attendance or compliance with the screening procedures is likely to occur. In the control arm, we will obtain data on contamination, that is, mostly self-initiated PSA testing.

When considering unforeseen lack of compliance with the protocol, all means to ensure objectivity in the exclusion principles from per-protocol analyses will be taken. Participants in both arms will be considered according to the same principles. Protocol deviations not related to the screening procedures are expected to appear in approximately 1:3 ratio for the arms. Obvious deviation from this ratio would be reported and interpreted as a potential source of bias.

## Blinding

Blinding in the conventional sense is not applicable: men are aware of being invited to screening. Hence, this is an open trial with screening and control arms.

Concrete measures to prevent bias, if any, from the awareness of the trial arm were nevertheless taken: (1) the control arm is blind to the fact that they are part of the trial; (2) allocation concealment is ensured by the centralised randomisation procedure preventing foreknowledge of upcoming arm allocation; and (3) communication to the general public on trial is kept to the minimum to prevent contamination (eg, by self-initiated PSA testing) among men in the control arm.

In addition, we underline that the primary outcome of the study, PCa death, is an objective outcome. The possibility of bias in its evaluation only relates to the assessment of the cause of death. The death certificates are filled by physicians with no involvement in the trial and can be assumed to be independent of trial arm, especially as deaths from PCa are likely to occur years after the diagnosis and hence unaffected by detection through screening or other means. Importantly, a previous study within the European Randomised Study of Screening for Prostate Cancer trial has shown that the cause-of-death data provided by Statistics Finland agreed almost perfectly with the assessment of a blinded expert panel in the Finnish centre of the trial and was independent of the trial arm.[17 18]

## Data collection process

Table 1 summarises the stages of the data collection process, targeted participants, and information and samples obtained.

## Study outcomes and other relevant variables

The primary outcome of the trial is death from PCa. Causes of death will be obtained from the Statistics Finland database and the underlying causes of death

**Table 1** Data collection process of the ProScreen trial

| Process stage | Target population | Information collected | Samples collected |
|---|---|---|---|
| Baseline | Participants | Family history<br>Previous PSA and Bx<br>Generic QoL/utility (15D, EQ5D)<br>Out-of-pocket costs<br>PSA and four-kallikrein panel | Plasma<br>Serum<br>Whole blood |
| MRI | Men with PSA >3 ng/mL and kallikrein score >7.5% | PI-RADS score | Digital image |
| Biopsy | Screen-positive men | Post-biopsy symptoms (0, 30 days)<br>Targeted fusion biopsies: number of ROIs, number of biopsies, length of samples<br>Systemic biopsies: biopsy length, cancer length and Gleason score per sample, total length of samples, total length of cancer, portion of cancer, global Gleason score, portion of Gleason 4 or 5, perineural invasion | Urine<br>Stool<br>RNA, DNA cancer tissue and prostate tissue<br>Plasma<br>Serum<br>Whole blood |
| Cancer diagnosis | Men with prostate cancer | Disease-specific QoL (EPIC-26, MAX-PC)<br>Generic QoL/utility (EQ5D, 15D), out-of-pocket costs<br>Gleason/ISUP grade group, number of positive cores, length of cancer, treatment, TNM stage | |

Bx, prostate biopsy; EPIC-26, Expanded Prostate cancer Index Composite; ISUP, International Society for Urological Pathology; MAX-PC, Memorial Anxiety Scale for Prostate Cancer; PI-RADS, Prostate Imaging Reporting and Data System; PSA, prostate-specific antigen; QoL, quality of life; ROIs, regions of interest; TNM, tumour, node, metastases.

will be considered when evaluating if the man died from PCa or from other causes. Cancer cases in the entire trial population including the control arm and non-participants in the screening arm are identified from pathology databases of the two hospitals and through linkages to the Finnish Cancer Registry using the unique personal identification numbers assigned to all Finnish residents to ensure complete coverage and avoid duplicates (double-count).

Secondary outcomes are:
► Diagnosis of PCa (divided into clinically significant and insignificant).
► Progression to advanced PCa (biochemical relapse or progression to metastatic).
► Death due to any cause.
► Cost-effectiveness of screening.

Adverse outcome variables to monitor screening-related harms are:
► Overdiagnosis of clinically insignificant PCa.
► Quality of life impacts of screening and quality of life among men with PCa (Expanded Prostate cancer Index Composite (EPIC-26) instrument).
► PCa-related anxiety (Memorial Anxiety Scale for Prostate Cancer (MAX-PC) questionnaire).
► Complications from biopsy (PRECISION questionnaire).

## STATISTICAL ANALYSIS
The main analyses will rely on the intention-to-screen (ITS) principle and will include all randomised men in the two trial arms who were alive and eligible (free of PCa) at the date of randomisation. Those men who became ineligible between the date of randomisation and first screening invitation will remain in the ITS analysis set. Men who were ineligible at the time of randomisation, but recognised as such only after randomisation, will be excluded from the ITS analysis set.

Two-sided statistical tests will be used, and the overall significance level will be set at 5%. Corresponding p values will be accompanied with estimates of differences and their 95% CIs.

### Analysis of the primary outcome
The primary outcome of the trial is death from PCa. This is a superiority trial regarding the primary outcome and the comparisons between trial arms will be analysed and presented on this basis.

Those men who survived will be considered as right-censored observations at the time passed between the time of analysis and time of randomisation period. Those men who were lost to follow-up (eg, due to emigration) will be considered as censored at that particular time (eg, at emigration). Time to death, defined as the difference

between the date of death and date of randomisation, will be used as the event time for the analysis.

To evaluate differences between screening and control arms in PCa-specific mortality, Gray's test[19] for testing the null hypothesis of equality of cumulative incidence functions will be used. This test differs from the commonly used log-rank test in how competing risks of death are treated and is based on the subdistribution hazard of PCa cause of death.

The test will be complemented by reporting the number of PCa deaths, number of men at risk and estimated cumulative incidence functions for each trial arm over follow-up time. The arms will be compared in absolute risks (number needed to invite, ie, the inverse of the risk difference and number needed to diagnose per averted PCa death, that is, the ratio of excess incidence to mortality reduction), as well as relative measures of effect (HRs). Descriptive summaries will also be presented by trial centres and age group at randomisation.

### Secondary analyses of the primary outcome

Fine-Gray model for the subdistribution hazard will be used to conduct analyses adjusted for background factors for the ITS analysis set. Outcomes will be compared between age groups and trial centres, and in case of differences, analyses to control for trial centre and for age at randomisation (categorised as 50–54, 55–59, 60–65 years) will be conducted.

Per-protocol analyses excluding men with repeated non-attendance will be conducted. We will estimate the screening effect on PCa mortality among those with at least one attended screening round relative to the entire control arm, as well as among those with at least two attended screening rounds. Additional analyses to correct for contamination and non-compliance, that is, estimation of efficacy of screening under ideal circumstances, will be conducted by the method of Cuzick et al.[20]

Descriptive analyses to assess effect heterogeneity by centre, age group, education and socioeconomic position will be performed to complement per-protocol analyses. Additional analyses requested by external reviewers or editors in peer-review processes will also be done.

### Analysis of secondary outcomes
#### Diagnosis of PCa

The analysis of cumulative incidence of PCa by disease aggressiveness intends to assess screening impact on detection of clinically significant PCa (representing potential benefit through early treatment) and clinically insignificant PCa (indicating overdiagnosis). The intention is to assess the extent of detection of clinically significant PCa by screening relative to the control arm, and extent of overdiagnosis relative to the control arm. This will inform about the degree of accomplishing rationale of the trial, that is, detection of aggressive cases at least similar to that in PSA-based screening, while substantially decreasing the yield of low-risk cases. As screening advances the time

of diagnosis by several years (lead time), cumulative incidence will be used as the indicator of risk.

Disease aggressiveness will be defined by the International Society for Urological Pathology (ISUP) Gleason grade group. The analyses will be conducted separately for the detection of clinically significant (Gleason 7+ or ISUP 2+) and clinically insignificant (Gleason <7 or ISUP 1) PCa. In secondary analyses, alternative criteria for clinically significant PCa will also be employed including ISUP 3+ (Gleason 4+3 or higher), maximum length of cancer tissue in biopsy and number of biopsy cores with cancer.

Risk differences and ratios will be used to infer screening benefits and overdiagnosis compared with the control arm. Besides cumulative incidence, the ratio of aggressive to non-aggressive cases (or proportion of aggressive cancers out of all PCa) will also be reported.

Cumulative incidence for both outcomes will be estimated by trial arm. The overall PCa incidence combines screening benefits and harms and is thus regarded of minor importance in the interpretation of screening impact. Tabular presentations of age at diagnosis, disease stage and grade at diagnosis will be presented.

Both ITS (by allocation) and per-protocol (screening participants and non-participants) analyses will be conducted for each screening round. For screening participants, screen-detected and interval cases will be reported separately, and screen-detected cases will be broken down by those detected in targeted biopsies of MRI-positive lesions (screening protocol evaluated) and systematic biopsies in screen-negative men with PSA density >0.15 (safety measure to avoid missing clinically significant cases). Any cases detected in a random sample of screen-negative men invited to MRI (analyses to assess underlying prevalence of PCa) will also be reported separately. Analyses to evaluate an optimised screening algorithm will include exclusion of cases with PI-RADS score 3 and kallikrein score calculated also incorporating information on previous biopsies (ignored in the main analysis), as well as using higher cut-off values for PSA and the kallikrein score.

### Advanced PCa

The analysis of advanced PCa will compare the cumulative incidence of cancer progression, including metastasis and/or biochemical relapse developing after diagnosis and primary treatment, between the screening and control arms. The purpose of the analysis is to evaluate differences between the arms in the risk of developing a potentially lethal, advanced PCa.

The origin of the analysis will be the time of randomisation. Cumulative incidence rates will be estimated by the Kaplan-Meier method, and differences between trial arms will be estimated by Cox regression models adjusted by age at diagnosis.

### Death due to any cause

The analysis of all-cause mortality aims to show that the trials arms are comparable with each other and the

general male population in Finland. These analyses will not inform about the effectiveness of screening. Cumulative survival and mortality rates will be estimated by the Kaplan-Meier method, from time of randomisation, displayed with frequencies of events and men at risk by trial arm, and by age at randomisation.

This analysis will focus on the ITS analysis set.

### Cost-effectiveness

A cost-effectiveness analysis will be performed, incorporating cost data for both out-of-pocket estimated from surveys and service cost data collected from healthcare providers, as well mortality results (ITS analysis) and utilities based on repeated surveys with 15D and EQ5D instruments (on a random sample of participants). The comparator is no active screening, here represented by the control arm. The main outcome is the incremental cost-effectiveness ratio in terms of costs per quality-adjusted life-year.

A preliminary and exploratory cost-effectiveness study can be conducted after the last of the follow-up surveys have been returned, approximately at 3 years after the randomisation of the last man into the trial. We plan to undertake a full cost-effectiveness analysis around the time when the evidence on the effectiveness of screening regarding primary outcome has been obtained; this will most likely be near to the analysis at 15 years.

### Quality of life

These analyses aim to evaluate the short-term and long-term impacts of screening on generic quality of life as well as disease-specific quality of life among men with PCa. Two disease-specific questionnaires, EPIC-26 instrument and MAX-PC questionnaires, will be used to measure quality of life at 0, 6, 12 and 24 months from PCa diagnosis in both trial arms.

Standard scoring of the EPIC-26 instrument will be used. Summary statistics of the five key domains over time and by trial arm will be calculated to assess changes in quality of life of men with PCa from diagnosis onwards. Summary and domain-grouped scores will be analysed using applications of linear models (or their non-parametric counterparts, if needed) for repeated measures to evaluate differences in quality of life between the arms following PCa diagnosis. Analyses will be adjusted by age group and disease aggressiveness (but not by stage, which is assumed to mediate the effect of screening on quality of life through stage shift).

PCa-related anxiety is measured with the MAX-PC questionnaire.[21] Results will be presented as frequencies and percentages for total and subscale scores by trial arm.

Generic quality of life and utilities are evaluated using the 15D and EQ5D instruments as described in the Cost-effectiveness section.

### Analysis of adverse outcomes

In addition to detection of low-risk disease by screening as an indicator of overdiagnosis, adverse outcomes mainly relate to the harms due to biopsies. Adverse effects of prostate biopsy are monitored using the questionnaire developed for the PRECISION trial covering pain and other symptoms immediately after biopsy and at 30 days following biopsy. The number of biopsies, as well as the number (%) and type of complications among those with biopsies, will be reported.

### Interim analyses and data monitoring

The first analysis of PCa mortality will be conducted at 10 years and the final analysis at 15 years (ie, at median follow-up time of 10 or 15 years). As we do not intend to stop the trial at 10 years, these interim analyses will be considered as preliminary information. Interim analyses at 10 years will include also analyses of shorter-term benefits.

To control the overall type I error rate (5%) of the trial, we will employ the O'Brien-Fleming rule for alpha spending function. We set the amount of information at 0.5 at 10 years based on the expected numbers of PCa deaths. Thus, by implementation of the O'Brien-Fleming algorithm, the resulting significance level at 10-year interim analysis will be 0.0056, and at the 15-year final analysis 0.0444.

The analyses of secondary outcomes will not be used to infer about the overall effectiveness of screening. We will consider the analyses of these distinct process measures as individual tests rather than part of disjunction testing, in which case, precise interpretation but not multiplicity adjustment will be necessary.[22 23]

Analyses of secondary endpoints informing about the intermediate outcomes of process indicators including participation, cancer detection, validity and diagnostic performance of the tests in the entire population and subgroups (screened men, non-participants, men in the control arm) will be carried out at regular intervals, as sufficient data become available for evaluation. These will inform about potential need to modify the procedures. Side studies using the samples collected will be carried out to identify new indicators of PCa risk and prognosis.

An independent data monitoring committee (DMC) oversees the trial conduct, and its main task is to ensure safety of the participants. Safety in this context means that screening or screening procedures should not lead to unacceptable disadvantage for the participants in the light of screening benefits. This could take place if the screening intervention had materially worse performance in detecting clinically relevant PCa than anticipated, or substantially higher level of overdiagnosis. The DMC is given a report of the screening results initially every 6 months and after the first year every 12 months. The DMC can also request any additional information they regard as pertinent to their task. In case of concern, the DMC can recommend discontinuation of the trial; in practice, that would mean stopping recruitment and discontinuation of further screening procedures. In addition, they have a mandate to suggest modifications to the trial protocol.

## Handling of missing data

Extent of missing data will be described, for example, by presenting the number of individuals with missing values per variable.

For outcome variables relying on dates—dates of randomisation, censoring, diagnosis or death—incomplete dates will be imputed by 15 (in the case that the day variable was missing, but known month and year), and by 30/6 (in the case that only the year was known).

In case a substantial proportion of men (at least 5%) have missing data on one or more variables needed for the effectiveness analysis in question, multiple imputation methods will be used to demonstrate the robustness of findings.[24] Imputation models will include outcome variables and trial arm in addition to all variables relevant to the particular analysis. Final estimates will be derived by combining estimates and their SEs across data sets using Rubin's rules.

## Data management and quality assurance

REDCap database application is used for data management in the trial, covering all major data types from questionnaires and laboratory results to MRI findings, diagnoses and causes of death. REDCap allows access defined by two-factor authentication and flexible definition of user-specific functions and rights.

In REDCap, variable-specific parameters and predetermined options are used to prevent entering invalid data (eg, predefined values and acceptable ranges). All data are verified from the original data source and monitored monthly. Until the verification, data are saved as incomplete or unverified. Lead times between screening tests are monitored every 6–8 weeks. For the laboratory work (including sampling, processing and storing), each task has a protocol shared by the study centres. Any deviations from the sample-specific protocol are documented.

**Contributors** All authors approved the final version of this manuscript. Study concept and design—AA, AR, JN, JR, KN, TK and TT. Drafting of the manuscript—AA and JN. Critical revision of the manuscript for important intellectual content—AA, AR, JN, JR, KN, TK and TT.

**Funding** This work was supported by the Academy of Finland (grant number 311336), the Finnish Cancer Foundation, the Jane and Aatos Erkko Foundation, Competitive State Research Funding administered by Tampere University Hospital (grant number 9V02), and Päivikki and Sakari Sohlberg Foundation.

**Competing interests** AR declares receipt of lecture and consultation fees from Janssen and Orion.

**Patient and public involvement** Patients and/or the public were not involved in the design, or conduct, or reporting, or dissemination plans of this research.

**Patient consent for publication** Not required.

**Provenance and peer review** Not commissioned; externally peer reviewed.

**ORCID iD**
Jaakko Nevalainen http://orcid.org/0000-0001-6295-0245

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
