## [Reviewer comments · BMJ Open]

ARTICLE DETAILS

TITLE (PROVISIONAL)	Protocol summary and statistical analysis plan for the randomized trial of early detection of clinically significant prostate cancer (ProScreen)
AUTHORS	Nevalainen, Jaakko; Raitanen, Jani; Natunen, Kari; Kilpelainen, Tuomas; Rannikko, Antti; Tammela, Teuvo; Auvinen, Anssi

VERSION 1 – REVIEW

REVIEWER	Weitzer, Friedrich Medical University of Graz, Dep. of Radiology, Div. of Nuclear Medicine
REVIEW RETURNED	21-Jun-2023

GENERAL COMMENTS	Excellent and well-written protocol summary without any major comments.
---

REVIEWER	Grey, Alistair University College London Hospitals NHS Foundation Trust, Department of Urology
REVIEW RETURNED	01-Sep-2023

GENERAL COMMENTS	This well written paper details outlines a trial screening programme which would trigger modern diagnostic techniques including mpMRI and targeted biopsy. This is important given both the equivocal outcomes of the previous large screening studies as well as the huge advances in prostate cancer diagnostics since their design. Statistical peer review is encouraged but from a urological perspective the trial design is sound.
---

REVIEWER	Yabes, Jonathan University of Pittsburgh, Medicine
REVIEW RETURNED	27-Oct-2023

GENERAL COMMENTS	This manuscript presents the protocol summary and statistical analysis plan for the ProScreen trial, a population-based, randomized, multicenter, pragmatic study aimed at evaluating the effectiveness of a novel screening algorithm in reducing prostate cancer (PCa)-specific mortality over a 15-year period when compared to a control group. The screening algorithm comprises three levels of risk assessment involving PSA, kallikrein panel, and MRI. Secondary outcomes encompass PCa diagnosis, PCa progression to advanced disease, all-cause mortality, cost-effectiveness, overdiagnosis, quality of life among those with PCa, PCa-related anxiety, and biopsy complications. Overall, the manuscript is well-written, providing sufficient details on the trial
--

	design parameters and the statistical analysis plan. However, several areas require further clarification and justification.  - In the section on sample size calculations, it is not clearly stated how the estimated number of deaths in the control group was derived, and the rationale for selecting an effect size of 0.75 hazard ratio is not provided. - The unequal 1:3 allocation ratio is not adequately justified. A rationale for this allocation ratio should be included. - The term randomization in 'batches' needs clarification. If it refers to permuted block randomization, the block sizes and whether they are randomized should be specified. - There was no indication that randomization was stratified by any variable. Why was area of residence (ie, Helsinki or Tampere), center or age group not considered for stratification? - The estimands, especially in the context of handling intercurrent events (e.g., protocol violations and compliance issues), are not clearly defined. For guidance, please refer to the ICH E9 (R1) addendum (https://www.ema.europa.eu/en/documents/scientific-guideline/ich-e9-r1-addendum-estimands-sensitivity-analysis-clinical-trials-guideline-statistical-principles_en.pdf) - There seems to be an inconsistency in how randomized participants who are later found to be ineligible will be handled. The statistical plan intends to include these patients based on the intent-to-screen (ITS) principle, while the protocol deviations subsection suggests they may be excluded. - The manuscript lacks sufficient detail regarding the per-protocol analyses and compliance/contamination analyses. It is essential to specify how confounding and bias will be addressed in these analyses. - There is no clear plan for accounting for the development of multiple cancers in the analyses. - The manuscript does not address the issue of controlling for multiplicity in the context of secondary outcomes. A justification for not performing adjustments should be provided. - The development of PCa is a post-randomization event, which may introduce confounding in the analysis of quality of life (QoL). The analysis plan should address this potential confounding. - A clear description of what patients in the control arm will receive is missing.
--	--

VERSION 1 – AUTHOR RESPONSE

Reviewer: 1

Dr. Friedrich Weitzer, Medical University of Graz

Comments: Excellent and well-written protocol summary without any major comments.

Response: We thank you for your positive assessment.

Reviewer: 2

Mr. Alistair Grey, University College London Hospitals NHS Foundation Trust

Comments: This well written paper details outlines a trial screening programme which would trigger modern diagnostic techniques including mpMRI and targeted biopsy. This is important given both the equivocal outcomes of the previous large screening studies as well as the huge advances in prostate cancer diagnostics since their design. Statistical peer review is encouraged but from a urological perspective the trial design is sound.

Response: We thank you for your positive assessment.

Reviewer: 3

Dr. Jonathan Yabes, University of Pittsburgh

Comments: This manuscript presents the protocol summary and statistical analysis plan for the ProScreen trial, a population-based, randomized, multicenter, pragmatic study aimed at evaluating the effectiveness of a novel screening algorithm in reducing prostate cancer (PCa)-specific mortality over a 15-year period when compared to a control group. The screening algorithm comprises three levels of risk assessment involving PSA, kallikrein panel, and MRI. Secondary outcomes encompass PCa diagnosis, PCa progression to advanced disease, all-cause mortality, cost-effectiveness, overdiagnosis, quality of life among those with PCa, PCa-related anxiety, and biopsy complications. Overall, the manuscript is well-written, providing sufficient details on the trial design parameters and the statistical analysis plan. However, several areas require further clarification and justification.

Response: We thank you for your encouraging general feedback as well as for your careful evaluation of the paper. We have made further clarifications to the paper as suggested and our point-by-point responses can be found below.

Comment: In the section on sample size calculations, it is not clearly stated how the estimated number of deaths in the control group was derived, and the rationale for selecting an effect size of 0.75 hazard ratio is not provided.

Response: We have modified the text to clarify this logic:

“With a 1:3 random allocation to the screening arm relative to the control arm, we expected that we would follow-up on 82 500 men in the control arm. Based on population statistics and causes of death registries, for the first ten years of the trial, we estimated PCa mortality to be 50 deaths per 100,000 person-years. For the entire 15 years follow-up, our estimate was 67 deaths per 100,000 person-years. Given deaths due to other causes, we thus expected to include at least 240 and 520 PCa deaths in the control arm during the first 10 and 15 years of the trial, respectively.”

Prior evidence has shown a 0.8 rate ratio between a PSA-based screening arm and a control arm in PCa mortality (Hugosson et al. 2019). Furthermore, there are indications that the addition of MRI and 4K to the screening protocol would improve the ability to detect clinically significant cancers (sensitivity 1.3-1.4 fold higher than TRUS-based biopsy, Drost Eur Urol 2020 PMID 31326219). Therefore, we anticipate a slightly larger (1.2-fold) screening benefit in PCa mortality in this trial than

in the previous PSA-based screening trials with TRUS-based based (above all ERSPC). We have added the text to the manuscript as well.

Comment: The unequal 1:3 allocation ratio is not adequately justified. A rationale for this allocation ratio should be included.

Response: We have added the following text to the manuscript to address this point:

“The rationale for the unequal randomization arose from the logistics in the conduct of the trial: we first estimated the maximum number of men that could be screened with existing resources. Given that the size of the control arm that does not receive any intervention has hardly any resource implications, we decided on an allocation ratio that would yield a manageable size of the intervention arm. Hence, the underlying rationale was to fix the size of the screening and maximize the size of the control arm to optimize the power of the study given the context (source population).”

Comment: The term randomization in 'batches' needs clarification. If it refers to permuted block randomization, the block sizes and whether they are randomized should be specified.

Response: The term refers to randomization of the population gradually a subgroup at a time and expansion of the randomization list. The lists of eligible men were constructed at different points of time during the enrolment phase to ensure that the time between randomization and the invitation would not be excessive. A long time could have resulted in problems with eligibility criteria (e.g., a man could have died or moved outside the study area between randomization and invitation). We have now indicated this in the text.

Comment: There was no indication that randomization was stratified by any variable. Why was area of residence (ie, Helsinki or Tampere), center or age group not considered for stratification?

Response: No stratification was used; the size of the trial is so large that we believe that a simple randomization guarantees a balance in key variables.

Comment: The estimands, especially in the context of handling intercurrent events (e.g., protocol violations and compliance issues), are not clearly defined. For guidance, please refer to the ICH E9 (R1) addendum (https://www.ema.europa.eu/en/documents/scientific-guideline/ich-e9-r1-addendum-estimands-sensitivity-analysis-clinical-trials-guideline-statistical-principles_en.pdf)

Response: Thank you for pointing out this shortcoming of the plan. We have now expanded and the rephased the text to provide clearer estimands for these analyses:

“Per protocol analyses excluding men with repeated non-attendance will be conducted. We will estimate the screening effect on PrCa mortality among those with at least one attended screening

round relative to the entire control arm, as well as among those with at least two attended screening rounds. Additional analyses to correct for contamination and non-compliance, i.e., estimation of efficacy of screening under ideal circumstances, will be conducted by the method of Cuzick et al. (1997).”

Comment: There seems to be an inconsistency in how randomized participants who are later found to be ineligible will be handled. The statistical plan intends to include these patients based on the intent-to-screen (ITS) principle, while the protocol deviations subsection suggests they may be excluded.

Response: The reviewer is correct that this was not consistently elaborated in the text. We intend not to include men in the ITS set who were not eligible at the time of randomization, but recognized as such only after randomization. Those who became ineligible between randomization and the invitation to screening, however, will be included in the ITS set. We have clarified this in the text.

Comment: The manuscript lacks sufficient detail regarding the per-protocol analyses and compliance/contamination analyses. It is essential to specify how confounding and bias will be addressed in these analyses.

Response: We have now clarified the estimands for these secondary analyses. While the ITS analysis can be considered to be free of confounding and bias, secondary analyses could be affected by healthy screenee bias (a form of selection bias): among the invited men, screening participants may differ from those not attending screening. To partially address this phenomenon, the analyses will consider education and socioeconomic position as control factors and this has been added to the text.

Comment: There is no clear plan for accounting for the development of multiple cancers in the analyses.

Response: As the end-point of interest is prostate cancer (diagnosis and death), only the first occurrence is relevant (disease recurrence is not included in either incidence or mortality) and there is phenomenon akin to e.g. a contralateral breast cancer. Recurrence of PCa would be a distinct entity; in our trial a healthy man can become of PCa patient only once, and regardless of disease events or treatment following diagnosis, we evaluate the effectiveness of screening via PCa mortality. We do not care whether a prostate cancer is the first primary cancer or not (i.e. it is assumed an independent event).

Comment: The manuscript does not address the issue of controlling for multiplicity in the context of secondary outcomes. A justification for not performing adjustments should be provided.

Response: We have added the following text to address this point:

“The analyses of secondary outcomes will not be used to infer about the overall effectiveness of screening. We will consider the analyses of these distinct process measures as individual tests rather than part of disjunction testing, in which case precise interpretation but not multiplicity adjustment will be necessary (Parker and Weir, 2022; Rubin, 2021).”

Parker, R.A., Weir, C.J. Multiple secondary outcome analyses: precise interpretation is important. *Trials* **23**, 27 (2022). <https://doi.org/10.1186/s13063-021-05975-2>

Rubin, M. When to adjust alpha during multiple testing: a consideration of disjunction, conjunction, and individual testing. *Synthese* **199**, 10969–11000 (2021). <https://doi.org/10.1007/s11229-021-03276-4>

Comment: The development of PCa is a post-randomization event, which may introduce confounding in the analysis of quality of life (QoL). The analysis plan should address this potential confounding.

Response: We agree with the reviewer that PCa diagnosis will affect QoL, especially as we use prostate cancer -specific QoL indicators. Therefore, we intend to adjust the analyses by age at diagnosis and disease aggressiveness (clinically significant or insignificant), which are features of prostate cancer diagnosis affected by screening, to control the confounding, and have added this to the manuscript. However, cancer stage will not be adjusted for as we consider it to be a factor through which the benefits of screening (earlier detection) are produced. The use of quality-adjusted life years will account for the longer time with prostate cancer diagnosis due to earlier detection (lead-time).

Comment: A clear description of what patients in the control arm will receive is missing.

Response: We have added the following text to the manuscript to clarify this:

“The control arm is unaware of being part of the trial, as they are not contacted nor subject to any trial procedures. At prostate cancer diagnosis, the aim is that all men in the control arm will also answer questionnaires, but otherwise they are treated according to standard clinical practice according to the well-established national treatment guidelines throughout the trial.”

Editor(s)' Comments to Author (if any):

Comment: Please reformat the main text so that it follows the structure recommended in the journal's instructions for authors for study protocols, for example the main text of your manuscript should contain an Ethics and Dissemination section. See: <https://bmjopen.bmj.com/pages/authors/#protocol>

Response: This heading has been added to the relevant part of the text.

Comment: Please remove the Conclusions section as this is not a requirement of BMJ Open.

Response: We have removed this section.

Comment: Along with your revised manuscript, please provide an example of the participant consent form as a 'Supplemental Material' file, as per item #32 of the SPIRIT checklist.

Response: We have now attached the consent form, which is only available in Finnish.

Comment: Along with your revised manuscript, please include a copy of the SPIRIT checklist indicating the page/line numbers of your manuscript where the relevant information can be found (<http://www.spirit-statement.org/>)

Response: The checklist is attached to the revision of the manuscript.

VERSION 2 – REVIEW

REVIEWER	Yabes, Jonathan University of Pittsburgh, Medicine
REVIEW RETURNED	12-Dec-2023
GENERAL COMMENTS	This revised manuscript adequately addressed areas of the study design and statistical analyses plan that needed further clarification and justification.